

# Transcriptomic and coexpression network analyses revealed the regulatory mechanism of *Cydia pomonella* infestation on the synthesis of phytohormones in walnut husks

Xiaoyan Cao[1], Xiaoqin Ye[2] and Adil Sattar[2]

[1] College of Horticulture, Xinjiang Agriculture University, Urumqi, China
[2] College of Forestry and Landscape Architecture, Xinjiang Agriculture University, Urumqi, China

Corresponding author
Adil Sattar, adl1968@126.com

## ABSTRACT

The codling moth (*Cydia pomonella*) has a major effect on the quality and yield of walnut fruit. Plant defences respond to insect infestation by activating hormonal signalling and the flavonoid biosynthetic pathway. However, little is known about the role of walnut husk hormones and flavonoid biosynthesis in response to *C. pomonella* infestation. The phytohormone content assay revealed that the contents of salicylic acid (SA), abscisic acid (ABA), jasmonic acid (JA), jasmonic acid-isoleucine conjugate (JA-ILE), jasmonic acid-valine (JA-Val) and methyl jasmonate (MeJA) increased after feeding at different time points (0, 12, 24, 36, 48, and 72 h) of walnut husk. RNA-seq analysis of walnut husks following *C. pomonella* feeding revealed a temporal pattern in differentially expressed genes (DEGs), with the number increasing from 3,988 at 12 h to 5,929 at 72 h postfeeding compared with the control at 0 h postfeeding. Walnut husks exhibited significant upregulation of genes involved in various defence pathways, including flavonoid biosynthesis (PAL, CYP73A, 4CL, CHS, CHI, F3H, ANS, and LAR), SA (PAL), ABA (ZEP and ABA2), and JA (AOS, AOC, OPR, JAZ, and MYC2) pathways. Three gene coexpression networks that had a significant positive association with these hormonal changes were constructed based on the basis of weighted gene coexpression network analysis (WGCNA). We identified several hub transcription factors, including the turquoise module (AIL6, MYB4, PRE6, WRKY71, WRKY31, ERF003, and WRKY75), the green module (bHLH79, PCL1, APRR5, ABI5, and ILR3), and the magenta module (ERF27, bHLH35, bHLH18, TIFY5A, WRKY31, and MYB44). Taken together, these findings provide useful genetic resources for exploring the defence response mediated by phytohormones in walnut husks.

## INTRODUCTION

Walnut (*Juglans regia*) belongs to the Juglandaceae family and is a perennial deciduous tree that is widely distributed worldwide and is rich in nutrients such as fat, protein and

phenolic substances (*Moser, 2012*; *Leslie et al., 2015*). According to statistics from 2018, the walnut cultivation area in China is approximately 487,000 ha, and the area cultivated in the Xinjiang Uyghur Autonomous Region accounts for nearly three-quarters of the total walnut cultivation area. However, the total production of walnuts in China is 1,785,000 t, which is 47.65% of the total global production. The Xinjiang Uygur Autonomous Region accounts for approximately 40% of the walnuts produced nationwide (*Ma et al., 2019*). However, the damage caused by insects to walnut production and quality has declined significantly in the Xinjiang Uyghur Autonomous Region.

The codling moth (*Cydia pomonella*) is a pest of pome fruit (apples and pears) worldwide and is highly destructive, adaptable, and stress resistant with high fecundity (*Chen & Dorn, 2010*). Notably, from 2016–2017, due to the *C. pomonella* outbreak in the apple orchards of Hotan Xinjiang, many apple trees were cut down, resulting in a lack of hosts. In 2018, it was discovered that there were many fallen fruits in the walnut orchards during the fruit maturity period. After quarantine, the insect pest responsible was found to be *C. pomonella*. For the first time, we discovered *C. pomonella* infestations in walnuts and an outbreak in Hotan, Xinjiang. This resulted in a 20–39% loss in walnut production, which has significantly limited the development of the walnut industry in Xinjiang (*Cao, Ye & Adil, 2023*). Walnut fruit is damaged by the larvae of *C. pomonella*, resulting in browning around the damaged area, as well as fruit rot and dropping (*Wang et al., 2019b*). At present, chemical control of *C. pomonella (*via *chemical pesticides)* is the main method used in pome fruit production. Although chemical pesticides have been used to control *C. pomonella*, they have not had marked effects. Chemical pesticides have been used to control *C. pomonella* Moreover, the use of chemical pesticides also causes environmental pollution (*Ju et al., 2021*). The breeding of resistant walnut plant cultivars is a good strategy for combating pest infestations, but the molecular mechanisms involved in the defence of walnut plants against *C. pomonella* damage are still unclear.

In nature, plants are constantly challenged by a variety of insects, including members of Coleoptera, Lepidoptera, and Hymenoptera. To survive or fend off attacks, plants have evolved many defence systems to resist insect attacks during long-term plant–insect interactions (*Acevedo et al., 2015*). The direct defence system of plants inhibits insect feeding by producing toxic compounds (*Chen, 2008*; *Howe & Jander, 2008*). For example, flavonoids are induced in poplar (*Populus*) in response to gypsy moth larval (*Lymantria dispar*) herbivory, which affects the duration of the developmental period and the survival rate of gypsy moth larvae (*Wang et al., 2019c*). Furthermore, flavonoids in ginkgo leaves have negative anti-herbivory effects on the fall webworm (*Hyphantria cunea*) (*Pan et al., 2016*). Our previous study revealed that the flavonoid content in walnut fruit increased after *C. pomonella* infestation, indicating that flavonoids may be involved in the defence of walnut plants against *C. pomonella* (*Cao, Ye & Adil, 2023*). The defence system of plants is mediated by phytohormones, including but not limited to jasmonate (JA), salicylic acid (SA), and abscisic acid (ABA) signalling (*Tzin et al., 2015*; *Thaler, Humphrey & Whiteman, 2012*). Plant receptors recognize insect feeding *via* elicitors, which may originate from insect saliva or even be products of endosymbiotic bacteria (*Chaudhary et al., 2014*; *Jiang et al., 2019*). Upon insect infection, a series of defence responses occur through the
recognition of insect elicitors, in which calcium- and ROS-related signalling plays an important role in triggering the activation of defence pathways such as the JA, SA, and ABA pathways. However, the accumulation of JA, SA and ABA plays an important role in regulating plant-induced resistance to insects (*Kerchev et al., 2012*; *Qiao et al., 2023*; *Wang et al., 2019a*; *Kawazu et al., 2012*).

The JA signalling pathway is involved mainly in plant defence against herbivorous insects and some pathogens (*Howe & Jander, 2008*; *Fürstenberg-Hägg, Zagrobelny & Bak, 2013*; *Machado et al., 2016*). In plant defence responses, the importance of JA signalling has been demonstrated, and a lack of JA biosynthesis or signalling results in severely hindered defence responses (*Kessler, Halitschke & Baldwin, 2004*; *Wang et al., 2008*). At the transcriptional level, transcriptional reprogramming due to mechanical damage or herbivory involves JA-signalling mediated (*Reymond et al., 2004*). For example, the defence response related to the JA signalling pathway is induced by insect infestation in maize and rice (*Qi et al., 2018*). Among them, the core components of the JA signalling pathway include mainly *LOX*, *AOS*, *OPR*, *JAZ*, *etc.*, and the transcription factors downstream of *JAZ*, such as *MYCs* (*Griffiths, 2020*). In addition to *MYCs*, many transcription factors, such as *ORA59*, *ERF1*, *bHLH3/13/14/17*, and *JAM1/2/3*, have been identified as key regulators of JA-mediated defence responses, (*Zhu et al., 2011*; *Mao et al., 2016*; *Song et al., 2013*; *Sasaki-Sekimoto et al., 2013*). Salicylic acid (SA) plays an important role in plant innate immunity, and its synthesis occurs *via* the isochorismate and phenylalanine ammonia-lyase pathways in plants (*Chen et al., 2009*; *Dempsey et al., 2011*). The SA signalling pathway is commonly involved in plant defence against piercing-sucking insects and saprophytic pathogens (*Pieterse et al., 2012*). For example, SA-induced defence responses are effective against sedentary sucking insects, such as aphids (*Zhang, Xue & Zhao, 2015*). Furthermore, some insects carry viruses or microorganisms that can trigger SA accumulation. *Tomato spotted wilt virus* transmitted by thrips feeding increases SA concentrations in Arabidopsis, leading to enhanced production performance and a preference of thrips for infected plants (*Abe et al., 2012*). Flagellin from *Pseudomonas* sp., present in the mouth of *L. decemlineata*, can induce SA accumulation in tomato leaves after feeding, thereby inhibiting JA-dependent defences, such as protease inhibitors and polyphenol oxidase, as well as herbivory-induced resistance (*Chung et al., 2013*). Plants synthesize ABA *via* the carotenoid pathway (*Arc et al., 2013*). The dynamics, signal transduction and function of abscisic acid in plants were discussed and reviewed in detail by *Chen et al. (2020)*. At present, the biosynthesis of ABA has been extensively studied in Arabidopsis. The ABA signalling pathway is generally involved in the defence of plants against leaf-chewing herbivores such as caterpillars (*Chen et al., 2020*; *Fernández de Bobadilla et al., 2022*). ABA deficiency increases plant susceptibility to herbivory (*Dinh, Baldwin & Galis, 2013*). In addition, ABA is involved in signalling processes that induce JA-dependent defence responses in systemic tissues (*Vos et al., 2013*). ABA induces the expression of COI-dependent MYCs, which induce plant resistance to insects by regulating many wound/herbivory response genes, such as *VSP*s, *LOX*s, and glucosinolate biosynthesis genes (*Schweizer et al., 2013*). However, although phytohormone-mediated signalling pathways have been extensively studied in plant

defence against herbivorous insects, phytohormone-mediated responses to *C. pomonella* damage in walnut husks are still unclear. Analysis of this mechanism is crucial for formulating effective prevention and control strategies, and more targeted exploration is needed.

The walnut variety Zha 343 is characterized by vigorous growth and high, stable yield. It is also the main variety grown in Hotan, Xinjiang. *C. pomonella* is a major quarantine pest in China. In the present study, we investigated the effects of walnut husks on the growth and development of *C. pomonella* larvae. Furthermore, we clarified the changes in the levels of major defence-related phytohormones (ABA, SA, JA, JA-Ile, Ja-Val and MeJA) in walnut husk at different time points (0, 12, 24, 36, 48 and 72 h) after *C. pomonella* infection. Afterwards, based on the walnut husk transcriptomic data, we systematically analysed the DEGs and the changes in the expression of genes related to phytohormone biosynthesis and signal transduction at different time points after *C. pomonella* infestation, and analysed the expression patterns of key genes in the flavonoid pathway. Finally, three gene regulatory networks closely related to changes in phytohormone content were constructed through WGCNA, and several key hub genes were identified. The results provide new insights into the study of insect resistance in walnut fruit and lay the foundation for the elucidation of defence mechanisms and the breeding of new insect-resistant walnut varieties.

## MATERIALS AND METHODS

### Plant materials and insect rearing

The walnut variety 'Zha 343' was selected as the test material, which was grown in the experimental field of Hotan County, Xinjiang Uygur Autonomous Region (42°22′N, 84°48′E). Zha 343 is a famous local walnut variety in Hotan County with a long cultivation history. It is characterized by thin walnut husks, full kernel and high nutritional value. Walnut row spacing is 7 m, and the distance between individual plants is 5 m. Plants free of pests and diseases, with normal development, an open canopy and uniform lighting, were selected. All the walnut fruits used for the experiments were developmentally similar and healthy. During the key period of walnut fruit development, meteorological parameters were recorded using at weather station (Table S1).

In March 2022, mature *C. pomonella* larvae were collected from a walnut orchard in Hotan County, Hotan Prefecture, Xinjiang Uyghur Autonomous Region. The larvae were placed in a Petri dish with corrugated paper, and the Petri dishes were placed in an artificial climate incubator. After eclosion, mating pairs (one male and one female) were placed in insect rearing boxes, with one pair per box, for mating and egg laying, and the mating pairs were provided with 10% honey water. Once hatched, the resulting each larva was fed individually in a Petri dish (9 cm × 9 cm × 1 cm) and fed with kernels. The feeding conditions were as follows: temperature, 25 ± 1 °C; humidity, 60 ± 5%; and photoperiod, 16:8 (light:dark) (*Fan, Wennmann & Jehle, 2019*). The larvae were fed until the 4rd instar (fourth moult) was reached for subsequent experiments.
## Plant treatments

When the walnut fruit was at the hard-core stage (approximately 80 days after flowering), 4rd instar *C. pomonella* larvae were introduced onto the surface of the walnut fruit and allowed to feed freely. The timing starts when the larvae begin to eat. Damaged walnut husks from 1 cm$^2$ surrounding the initial *C. pomonella* feeding sites were taken from the walnut fruit with a knife (Fig. S1), and the insect droppings at the borehole were cleaned with pure water. Walnut husk samples were collected for gene expression and phytohormone analyses at 0, 12, 24, 36, 48, and 72 h after larval infestation. The plants at 0 h postinfestation were used as control plants, and walnut husk sections were taken as previously described. Walnut husks from three samples per time point were pooled for each biological replicate, and three repetitions were performed at each time point. All the samples were cut into small pieces, placed in centrifuge tubes, immediately frozen in liquid nitrogen, and stored at −80 °C. The infestations of *C. pomonella* were staggered to ensure that all the samples were collected at the same time to avoid the impact of different fruit development stages. The meteorological parameters for the collection date are shown in Table S1.

## Transcriptomic analysis of walnuts infested with larvae

The walnut husk samples were taken out from the −80 °C ultralow temperature freezer and immediately placed them in a foam box containing liquid nitrogen., The samples were subsequently removed from the foam box and ground into powder in liquid nitrogen with a sterilized mortar and pestle. The powdered samples were then immediately transferred into 1.5 ml RNase-free microtubes (Corning Incorporated, Corning, NY, USA). Afterwards, total RNA was isolated from each sample *via* TRIzol reagent according to the manufacturer's instructions (Invitrogen, Carlsbad, CA, USA). Transcriptome libraries were constructed *via* the NEBNext® Ultra™ RNA Library Prep Kit for Illumina® (NEB, USA). The sequencing libraries were sequenced on the Illumina NovaSeq 6000 platform. The raw data were processed *via* fastp v 0.19.3, and the reference genomes and their annotation files were downloaded from NCBI (*Martínez-García et al., 2016*). HISAT v2.1.0 was used to construct the index and compare the clean reads to the reference genome. RNA-seq data analysis was performed according to the protocols of previous studies (*Conesa et al., 2016*). The expression values of all genes were calculated and normalized to fragments per kilobase of transcript per million fragments mapped (FPKM). The correlation between different samples was analysed *via* principal component analysis (PCA), which was implemented in the MetWare online tool (https://cloud.metware.cn/#/tools/tool-list). DEG analysis was performed on six groups (three biological replicates each) *via* the DESeq R package (1.18.0). The *p* value was corrected *via* the method of *Love, Huber & Anders (2014)*. The DEGs were identified by an absolute value of log2(fold change) ≥1 and a false discovery rate (FDR) <0.05. The statistical power of this experimental design, calculated in RNASeqPower, is 0.86. Additionally, heatmaps of selected genes were generated *via* the MetWare online tool (https://cloud.metware.cn/#/tools/tool-list).

## KEGG analysis of DEGs and identification of co-expression modules

The enrichment of DEGs in the Kyoto Encyclopedia of Genes and Genomes (KEGG) pathways was analysed *via* KOBAS (*Kanehisa et al., 2008*). On the basis of the fragments per kilobase of transcript per million mapped reads (FPKM) data, the R package WGCNA was used to identify highly correlated gene modules (*Langfelder & Horvath, 2008*; *Zhan et al., 2015*). The R package DCGL was used to screen genes on the basis of gene expression and variation, and the TOM Similarity algorithm was used to convert the adjacency matrix into a topological overlap (TO) matrix. Modules with highly correlated eigengenes (correlation >0.8) were merged.

## Visualization of the hub genes

Genes that have the highest degree of intramodule connectivity are called intramodule hub genes (*Langfelder & Horvath, 2008*). In the present study, the top 200 genes were defined as hub genes according to k {$k_{cor,\ i}^{(q)} = cor[xi,\ ^{E(q)}]$} and ME (module eigengene). The hub genes of each module were compared, and the top 200 hub genes of each module were visualized *via* Cytoscape. The gene annotation information was obtained from the KOBAS 2.0 annotation results.

## QUANTITATIVE STUDY OF PHYTOHORMONES

The quantification of abscisic acid (ABA), salicylic acid (SA), jasmonic acid (JA), JA-isoleucine conjugate (JA-Ile), methyl jasmonate (MeJA) and jasmonic acid-valine (JA-Val) from walnut husks was performed *via* high-performance liquid chromatography (HPLC) mass spectrometry (*Pan, Welti & Wang, 2010*). The determination of phytohormone levels was carried out *via* MetWare (http://www.metware.cn/) on the AB Sciex QTRAP 6500 LC-MS/MS platform (https://sciex.com.cn/). Each treatment was replicated three times.

## Quantitative real-time PCR (qRT–PCR) analysis

qRT-PCR analysis was performed on an Applied Biosystems 7500 Fast Real-Time PCR System (Applied Biosystems, Foster City, CA) using Hifair® III 1st Strand cDNA Synthesis SuperMix (YEASEN). The PCR primers were designed *via* NCBI Primer-BLAST (https://www.ncbi.nlm.nih.gov/tools/primer-blast/) and are listed in Table S2. The volume of the qRT-PCR amplification mixture was 20 µL, and the mixture included sterile ultrapure water (7 µL), forward primer (10 µM) (0.5 µL), reverse primer (10 µM) (0.5 µL), DNA template (2 µL), and Hieff® qPCR SYBR Green Master Mix (Low Rox Plus) (10 µL). All the experiments were performed on ice. The amplification program was 95 °C for 10 s, 60 °C for 30 s, and 72 °C for 10 s for a total of 40 cycles. The relative gene expression was calculated *via* the $2^{-\Delta\Delta Ct}$ method (*Livak & Schmittgen, 2001*).

## Data analysis

All the data are presented as the means of several values. The difference in phytohormone levels in walnut husks at different time of *C. pomonella* infestation were analysed *via*

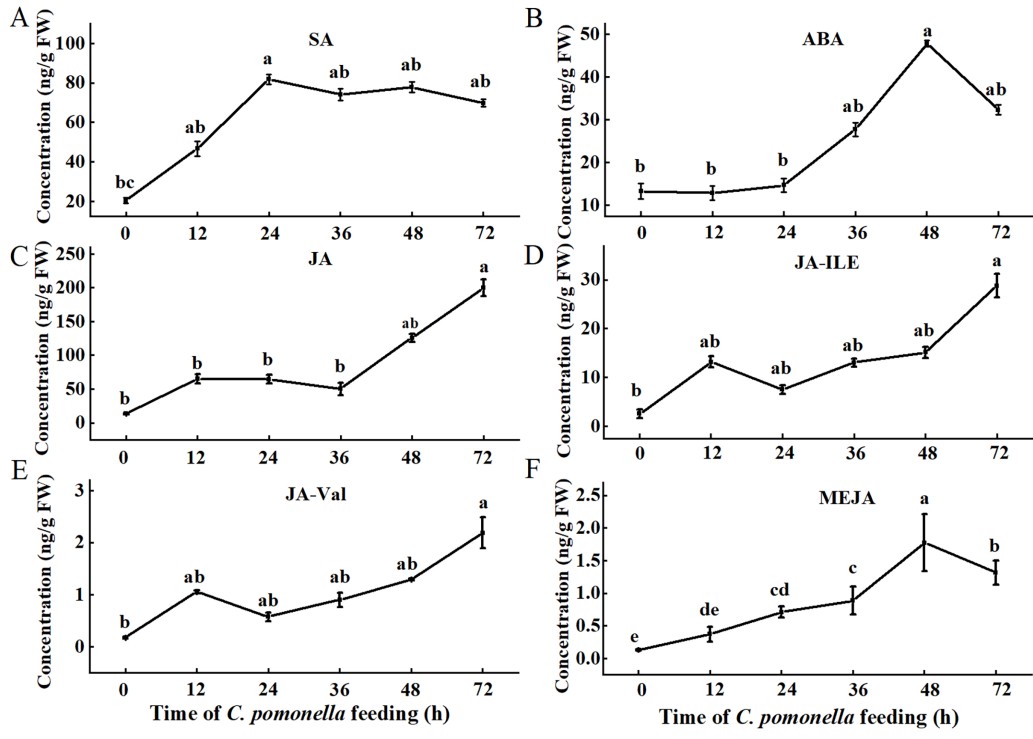

**Figure 1 Changes in phytohormone content at different times during *C. pomonella* infestation.** Changes in the SA (A), ABA (B), JA (C), JA-Ile (D), JA-Val (E), and MeJA (F) levels in walnut husk. The data represent the mean ± SE, and different lowercase letters indicate significant differences ($P < 0.05$).

one-way analysis of variance (ANOVA) by IBM SPSS 23.0 (IBM SPSS Software Inc., Chicago, IL, USA). Means were considered statistically significant at $p < 0.05$.

# RESULTS

## Variation characteristics of phytohormone concentration

Since phytohormones are involved in plant defences against insect infestation, we tested the levels of several phytohormones during different periods. The trends of the SA, ABA, and MeJA levels initially increased but then subsequently decreased. The SA concentration peaked 24 h after *C. pomonella* infestation, and the levels of ABA and MeJA peaked 48 h after infestation (Figs. 1A, 1B, 1F). The concentrations of JA, JA-Ile, and JA-Val tended to increase within 72 h of infestation (Figs. 1C–1E).

## Transcriptomic analysis of the response of walnut husks to *C. pomonella* feeding

Our LC–MS/MS datasets reliably captured changes in phytohormones such as SA, ABA, and JA (Fig. 1). We next sought to analyse the effects of *C. pomonella* damage on walnut husk hormones at different times through transcriptome-based sequencing. A comparative analysis of the transcriptome of the walnut husks was performed at 0, 12, 24, 36, 48, and 72 h after infestation. The total number of clean reads in the RNA-seq library ranged from

41,402,492 to 48,358,932. After low-quality reads were removed, a total of 120.34 Gb of clean data was obtained for subsequent analysis. A total of 936 million reads were generated. Approximately 90% of the unique matches in the total number of clean reads matched the reference genome. The quality assessment of the transcriptome data revealed low error rates (0.03%), whereas the Q30 was very high (>91%) (Table S3).

The transcript level was analysed *via* the fragments per kilobase of transcript per million mapped reads (FPKM) value, and a total of 31,124 gene transcripts were detected in all the samples (Table S4). Gene expression levels for each replicate were assessed *via* principal component analysis (PCA) (Fig. S2). Using an absolute value of log2 (fold change) ≥1 and a FDR < 0.05 as the screening conditions for DEGs, a total of 11,662 DEGs were screened and further analysed. There were 3,988 (2,533 upregulated, 1,455 downregulated), 4,510 (3,063 upregulated, 1,447 downregulated), 3,357 (2,069 upregulated, 1,288 downregulated), 2,911 (1,971 upregulated, 940 downregulated), and 5,929 (3,703 upregulated, 2,226 downregulated) DEGs for the 12, 24, 36, 48, and 72 h samples, respectively (Fig. 2A, Tables S5 and S6). The distributions of upregulated and downregulated DEGs at the five time points were calculated and plotted in a Venn diagram. The expression of 691 DEGs was upregulated at 12 h after larval infestation and then returned to normal, and the expression of the other 674 DEGs was upregulated at 72 h (Fig. 2B).

The DEGs from each time point were further analysed *via* KEGG enrichment analysis to uncover the major pathways (Table S7). With $p <0.05$ as the screening criterion, DEGs were screened at 12, 24, 36, 48, and 72 h after infestation, and the results revealed 21, 29, 24, 19, and 28 pathways, respectively, from the KEGG database. Among these significant pathways, the biosynthesis of secondary metabolites, metabolic pathways, alpha-linolenic acid metabolism, and phenylpropanoid biosynthesis were involved in the response of the walnut husk to *C. pomonella* at all time points after infestation (Fig. 2C). The top 10 most important pathways at each time point were further summarized, and two pathways, biosynthesis of secondary metabolites and metabolic pathways, had the most enriched genes (Fig. 2D).

## Phytohormone and metabolite biosynthetic genes

Defence responses mediated by phytohormone signal transduction pathways play an important role in plant defence against herbivorous pests. Therefore, we further analysed the expression of key genes involved in the SA, ABA, and JA synthesis pathways. Phenylalanine ammonia lyase (PAL) is the major gene involved in the PAL pathway for SA biosynthesis. Five *PAL* genes were significantly upregulated. (Fig. 3A). Among the genes involved in ABA biosynthesis and signal transduction, the transcripts of zeaxanthin epoxidase (*ZEP*), xanthoxin dehydrogenase (*ABA2*), and the abscisic acid receptor PYL family (*PYL*) were significantly upregulated after *C. pomonella* feeding damage. Moreover, 9-cis-epoxytenoid dioxygenase (*NCED*) was significantly downregulated after *C. pomonella* feeding damage, whereas ABA-responsive element binding factor (*ABF*) was upregulated only at 36 and 48 h (Fig. 3B). Among the genes involved in JA biosynthesis and signal transduction, lipoxygenase (*LOX*), allene oxide synthase (*AOS*), allene oxide

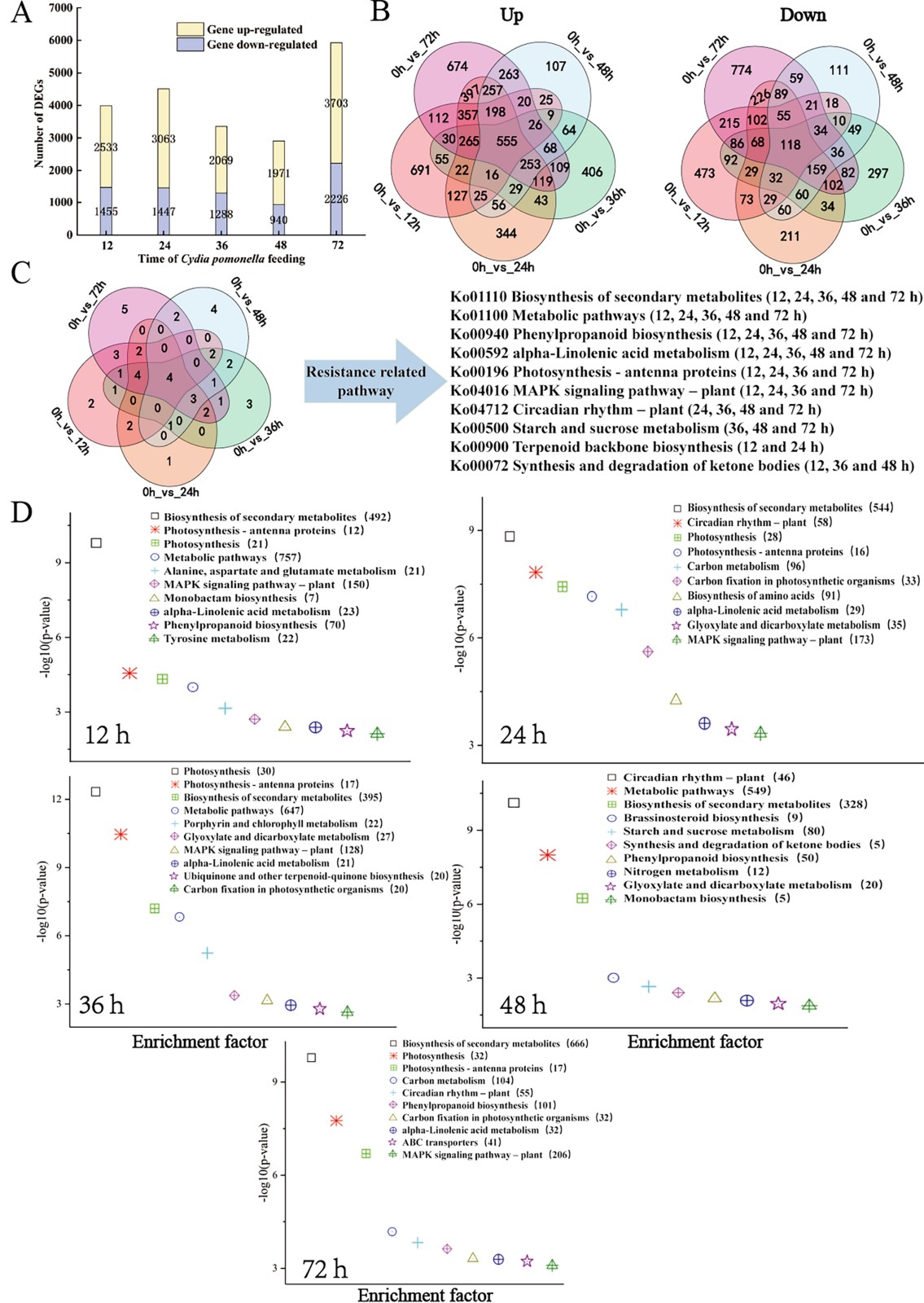

**Figure 2 Overview of the response of the walnut husk transcriptome to *C. pomonella* damage over time.** (A) Number of individual transcripts with significantly increased or decreased levels at each time point. (B) Venn diagram showing the number of transcripts whose levels increased or

**Figure 2** (continued)
decreased at different time points after *C. pomonella* infestation. (C) Venn diagram showing the overlap of shared and unique pathways in the transcriptome at 12, 24, 36, 48, and 72 h after *C. pomonella* infection. (D) KEGG pathway enrichment analysis of DEGs in the transcriptome of walnut husks at 12, 24, 36, 48, and 72 h after *C. pomonella* infestation. The data are displayed with a scatter plot, in which the *p* value represents the log10 (*p* value). The values in parentheses indicate the number of DEGs in each pathway.     

cyclase (*AOC*), 12-oxophytodienoate reductase (*OPR*), and the jasmonate-ZIM domain (*JAZ*) were significantly upregulated following *C. pomonella* infection. The transcription factor MYC2 is a global regulator of JA signalling (*Ogawa et al., 2017*); notably, only *MYC2* (LOC108991342) was significantly upregulated at 72 h compared with 0 h, and the other transcription factor *MYC*s were significantly upregulated at 72 h after *C. pomonella* infestation (Fig. 3C). Studies on genes related to flavonoid biosynthesis have shown that the transcript levels of phenylalanine ammonia-lyase (*PAL*), trans-cinnamate 4-monooxygenase (*CYP73A*), 4-coumarate-CoA ligase (*4CL*), chalcone synthase (*CHS*), chalcone isomerase (*CHI*), flavanone 3-hydroxylase (*F3H*), anthocyanidin synthase (*ANS*), and leucoanthocyanidin reductase (*LAR*) are significantly upregulated in walnut husks following *C. pomonella* damage. Three of these genes were involved in the synthesis of coumaroyl-CoA (the initial metabolite of flavonoid biosynthesis), and five genes are involved in the biosynthesis of flavonoids (Fig. 3D).

## Identification of gene coexpression modules *via* WGCNA

Weighted gene coexpression network analysis (WGCNA) is a method for identifying gene networks with related functions or traits and revealing putative hub genes with specific effects (*Langfelder & Horvath, 2008*). WGCNA was used to group genes sharing the same expression pattern. The 11,662 DEGs were clustered into 16 different modules (Fig. 4A, Table S8), and these modules are displayed in different colours. Using the levels of ABA, SA, JA, JA-Ile, JA-Val, and MeJA in the corresponding samples as the phenotypic data, the gene module–trait correlation was analysed, an intermodule correlation heatmap (Fig. 4B) was generated, and each branch of the tree corresponded to a group of highly related genes. There was a high degree of topological overlap among genes within the same module. The module that correlated strongly with ABA and MeJA was the green module (correlation value, cor = 0.61, *p* value = 0.0072), which contained 741 genes. The turquoise module was strongly correlated with SA (correlation value, cor = 0.58, *p* value = 0.012), which contained 2,440 genes. The module strongly correlated with JA, JA-Ile, and JA-Val was the magenta module (correlation value, cor = 0.60, *p* value = 0.0085); this module included 609 genes (Fig. 4C).

## Analysis of the turquoise, green, and magenta modules

Moreover, the KEGG database was used to explore the main biological pathways associated with the turquoise, green, and magenta modules. The DEGs in the turquoise module were enriched mainly in metabolic pathways, biosynthesis of secondary metabolites, and the MAPK signalling pathway (Fig. 5A). The metabolic pathways, photosynthesis, and circadian rhythm-plant pathways were enriched mainly in the DEGs

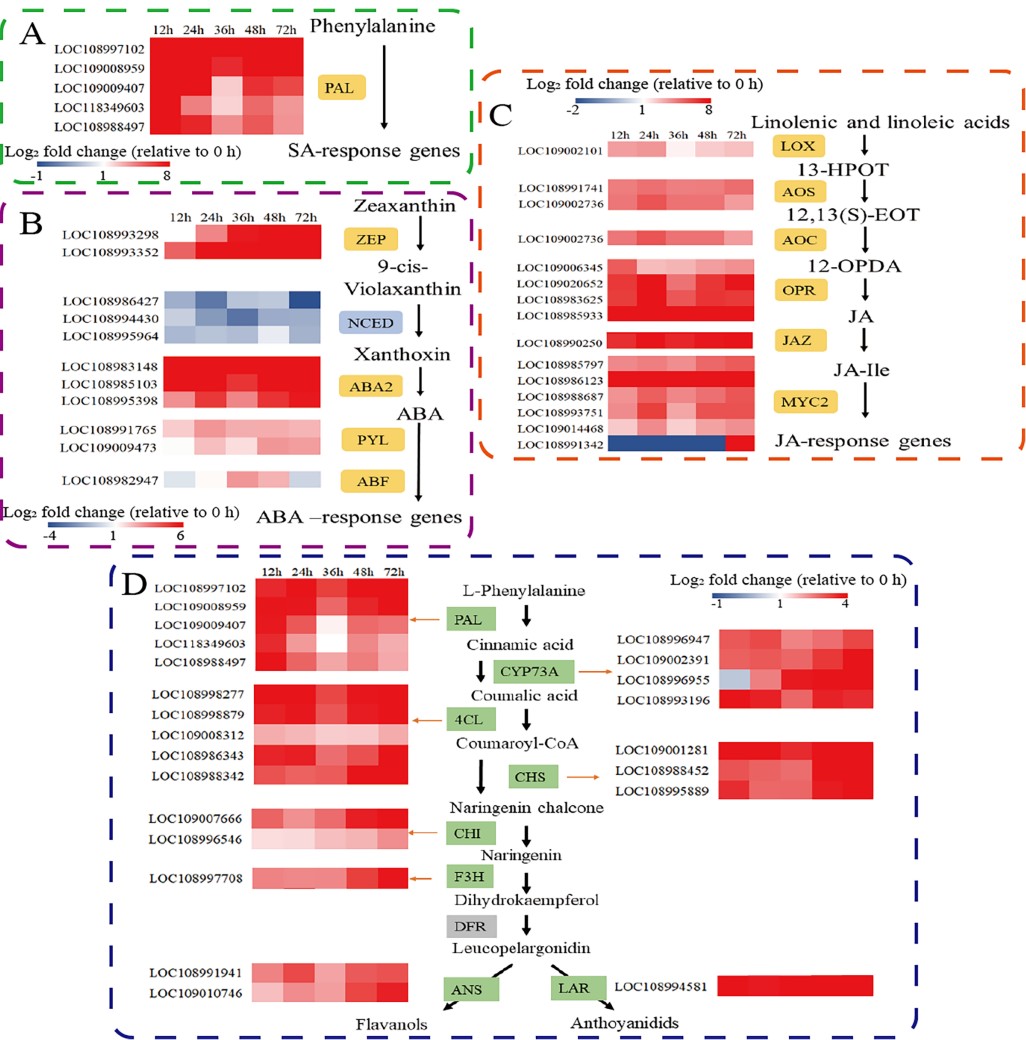

**Figure 3 Responses of *C. pomonella* to the salicylic acid (SA), abscisic acid (ABA), jasmonic acid (JA), and flavonoid signalling pathways.** The red and blue boxes indicate upregulated and down-regulated genes, respectively. PAL, phenylalanine ammonia lyase; ZEP, zeaxanthin epoxidase; NCED, 9-cis-epoxy carotenoid dioxygenase; ABA2, xanthoxin dehydrogenase; PYL, abscisic acid receptor PYL family; ABF, ABA-responsive element binding factor; LOX, lipoxygenase; 13-HPOT, 13(S)-hydro-peroxylinolenic acid; AOS, allene oxide synthase; 12,13-EOT, 12,13(S)-epoxy linolenic acid; AOC, allene oxide cyclase; 12-OPDA, 12-oxocis-10,15-phytodienoic acid; OPR, 12-oxophytodienoate reductase; JAZ, jasmonate-ZIM domain; JA-Ile, jasmonate-L-isoleucine; MYC2, transcription factor MYC2; CYP73A, trans-cinnamate 4-monooxygenase; 4CL, 4-coumarate-CoA ligase; CHS, chalcone synthase; CHI, chalcone isomerase; F3H, flavanone 3-hydroxylase; ANS, anthocyanidin synthase; LAR, leucoanthocyanidin reductase.

in the green module (Fig. 5B). The phytohormone signalling, ABC transporter, and alpha-linolenic acid metabolism pathways were enriched mainly in the DEGs in the magenta module (Fig. 5C). This finding is fundamentally consistent with the pathway enrichment of DEGs whose expression was upregulated. These findings indicate that these phytohormones may be involved in the walnut husk defence response against *C. pomonella* by regulating the expression of genes related to the abovementioned signalling pathways.

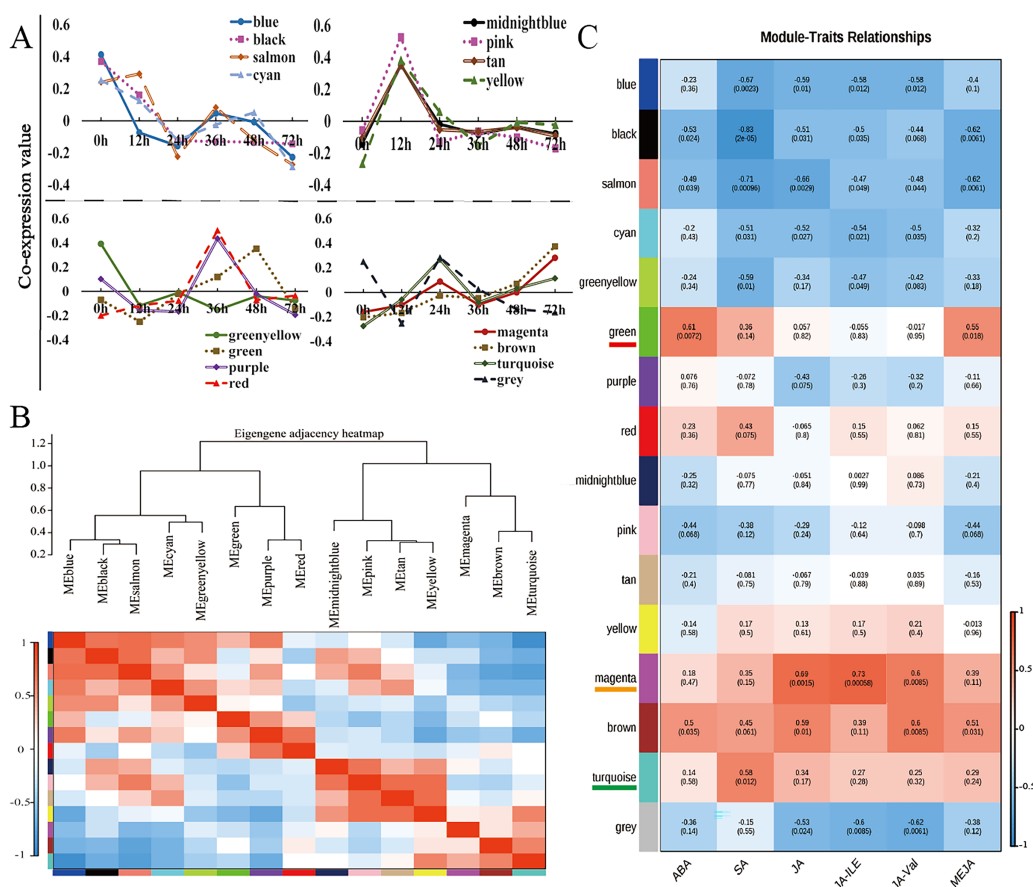

**Figure 4 WGCNA of the modules strongly correlated with ABA, SA, JA, JA-Ile, JA-Val, and MeJA.**
(A) Weighted gene correlation network analysis (WGCNA) of differentially expressed genes (DEGs) in walnut husks at seven different timepoints after *C. pomonella* infection using FPKM values. (B) Hierarchical clustering dendrogram and heatmap of related MEs. (C) Weight correlations and *p* values (in parentheses) for modules related to ABA, SA, JA, JA-Ile, JA-Val, and MeJA. Modules highly related to ABA and MeJA are underlined in red under the module name, modules highly related to SA are underlined in orange under the module name, and modules highly related to JA, JA-Ile, and JA-Val are underlined in green under the module name.

Among these genes, 94 in the turquoise module were related to the MAPK signalling pathway, and 32 in the magenta module were related to the phytohormone signalling pathway. The genes related to these two pathways were further analysed, and the heatmap for the cluster analysis results revealed that in the turquoise module, in response to codling larval herbivory, the number of genes in the MAPK signalling pathway gradually increased, and the expression of these genes was upregulated at 24 h. These results suggest that the genes in the MAPK signalling pathway may play an important role in the regulatory network of SA biosynthesis (Fig. 5D, Table S9). In the magenta module, the genes related to the phytohormone signalling pathway were analysed, and the genes related to jasmonic acid signalling were significantly enriched after *C. pomonella* damage, during which the expression level continued to increase. These genes are involved in the JA pathway and may play an important role in the regulatory network (Fig. 5E, Table S10).

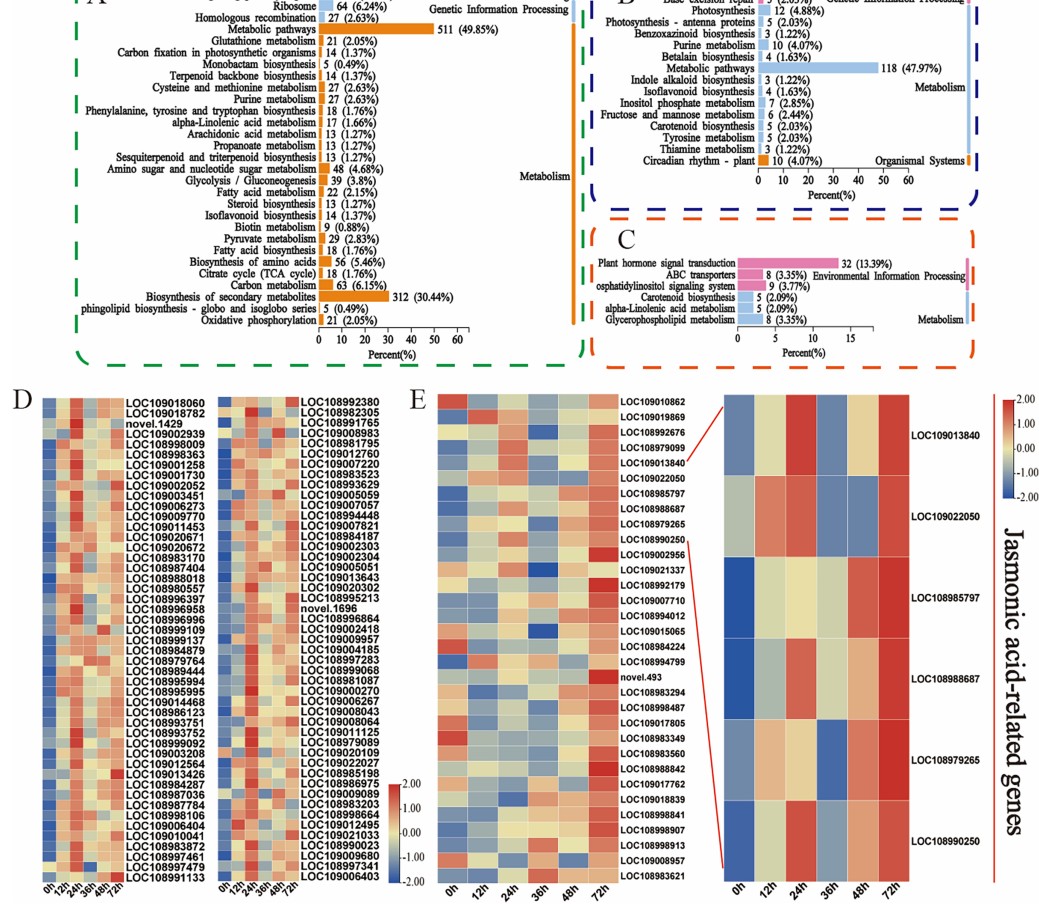

**Figure 5 Functional classification and comparison of DEGs in the turquoise, green, and magenta modules.** (A) KEGG pathway enrichment analysis of the DEGs in the turquoise module. (B) KEGG pathway enrichment analysis of the DEGs in the green module. (C) KEGG pathway enrichment analysis of the DEGs in the magenta module. (D) DEGs related to the MAPK signalling pathway in the turquoise module. (E) DEGs related to phytohormone signal transduction in the magenta module. The values are log2-fold change values of FPKM.

## Identification of the hub genes of each module

Hub genes are highly connected nodes in expression networks and are involved in many biological processes and interactions (*Zhang & Horvath, 2005*). We screened the top 200 genes with the highest correlation in the module as hub genes. As integral players in plant growth, transcription factors (TFs) regulate many target genes, thereby affecting biological processes such as plant insect resistance. Then, 15, 21, and 18 hub transcription factors in the turquoise, magenta and green modules, respectively, were screened out, and the expression of the key transcription factors was analysed (Table S11). In the turquoise module, the expression of transcription factors, including *AIL6* (LOC108979694), *MYB4* (LOC108991021), *PRE6* (LOC109019220), *WRKY71* (LOC108990221), *WRKY31* (LOC108984287), *ERF003* (LOC109019370), and *WRKY75* (LOC109003208), was the focus (Fig. 6A). The relative expression of these genes was significantly upregulated after *C. pomonella* infestation, and the range of upregulation levels was greatest at 24 h after

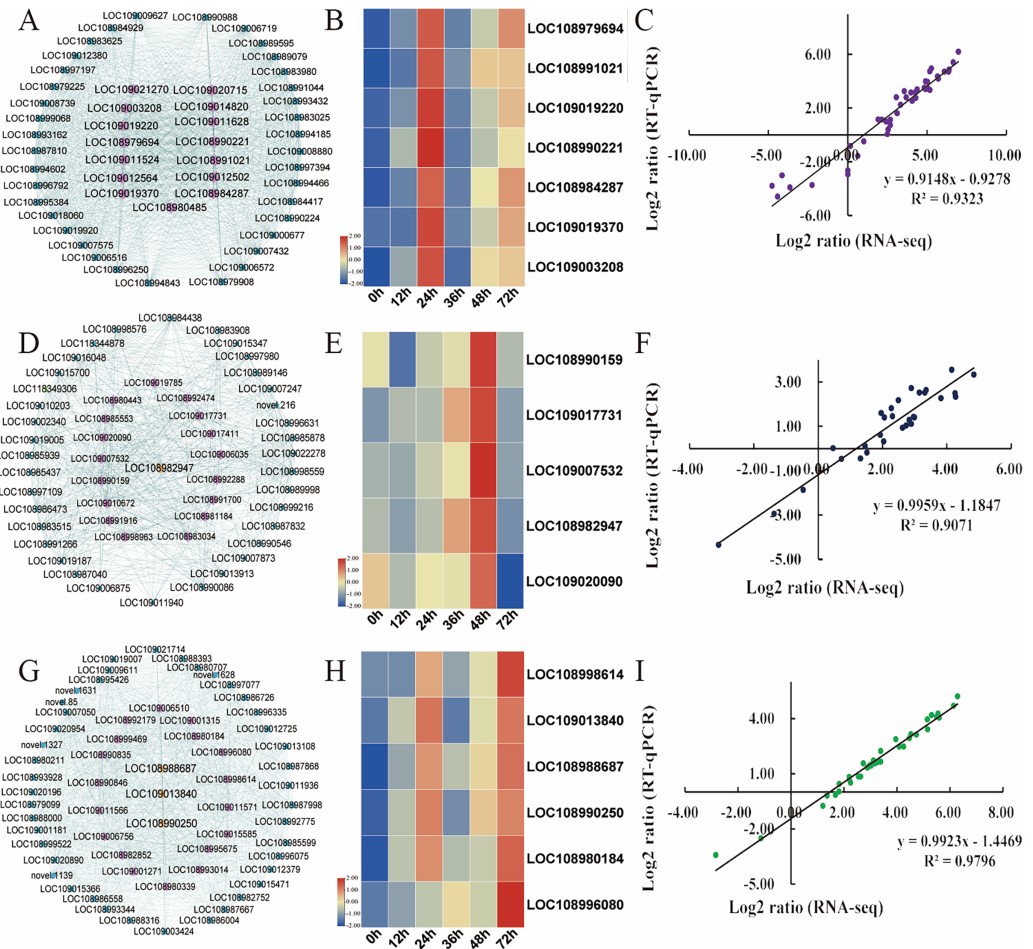

**Figure 6 Identification and selection of hub genes of each module.** (A) Hub genes and related gene networks in the turquoise module. (B) Heatmap depicting the expression profiles of seven transcription factors in the turquoise module. (C) Correlations of RNA-seq and qRT-PCR gene expression data in the turquoise module. (D) Hub genes and related gene networks in the green module. (E) Heatmap depicting the expression profiles of five transcription factors in the green module. (F) Correlations of RNA-seq and qRT-PCR gene expression data in the green module. (G) Hub genes and related gene networks in the magenta module. (H) Heatmap depicting the expression profiles of six transcription factors in the magenta module. (I) Correlations of RNA-seq and qRT-PCR gene expression data in the magenta module. According to the correlation of the RNA-seq and qRT-PCR gene expression data, the relative expression levels are shown as log2-fold change (FC) values.

infestation (Fig. 6B). In the green module, the transcription factors *bHLH79* (LOC108990159), *PCL1* (LOC109017731), *APRR5* (LOC109007532), *ABI5* (LOC108982947), and *ILR3* (LOC109020090) were the focus of interest (Fig. 6D). According to the expression pattern, the expression level was significantly upregulated at 48 h after *C. pomonella* infestation (Fig. 6E). The transcription factors we focused on in the magenta module included *ERF27* (LOC108998614), *bHLH35* (LOC109013840), *bHLH18* (LOC108988687), *TIFY5A* (LOC108990250), *WRKY31* (LOC108980184), and *MYB44* (LOC10899608) (Fig. 6G). Similar to the expression patterns of the TFs in the other two modules, the expression of the TFs of interest in the magenta module was significantly

upregulated at 72 h after *C. pomonella* infestation (Fig. 6H). To visualize the reliability of the RNA-seq data, qRT-PCR was used to analyse the expression levels of these 18 hub genes, and the correlation analysis of the RNA-seq and qRT-PCR data showed that the $R^2$ for each module was greater than 0.90. These results indicated that the hub genes were reliable (Figs. 6C, 6F, 6I). Furthermore, on the basis of the expression patterns of these genes, these genes are considered key candidate genes for the response of walnut husks to *C. pomonella* feeding and for the regulation of walnut shell defence against *C. pomonella* infestations.

## DISCUSSION

*C. pomonella* is a major global pest of pome fruits (apples and pears) and walnuts (*Schifani, Giannetti & Grasso, 2023*). *C. pomonella* begins to damage walnut fruits during the fruit enlargement period (30 days after flowering) and continues to damage walnut fruit throughout its developmental stages. Currently, research on *C. pomonella* in walnut orchards has focused mainly on its biological characteristics, detoxification, and control (*Curtiss et al., 2023*; *Piskorski & Dorn, 2011*; *Kadoić Balaško et al., 2020*). However, there have been few reports on the defence adaptations of *C. pomonella* host plants and their response mechanisms. In this study, we analysed the transcriptome dynamics of the husks of the field-grown walnut Zha343 cultivar following *C. pomonella* attack. At present, no research has reported the transcriptomic response of walnut husk at different time points of *C. pomonella* infestation during the hardcore stage of walnut fruits. In the present study, transcriptome analysis elucidated the response of walnut husk to *C. pomonella* infestation.

Phytohormones participate not only in the growth and development of plants but also in plant defences against biotic stress (*Gasperini & Howe, 2024*; *Singh et al., 2024*). The SA signalling pathway is triggered mainly by the piercing and sucking mouthparts of insects (*Zeng, Watanabe & Yang, 2019*). In the present study, we found that the SA concentration did not continually increase, but rather, first increased but then decreased, and the levels of ABA and MeJA exhibited a similar trend. These results are consistent with those of previous studies in which the levels of SA and ABA in maize first increased but then decreased after Asian corn borer (*Ostrinia furnacalis*) infestation (*Guo et al., 2019*). However, in a previous study of tea trees, an increase in the SA content in response to herbivory by the tea geometrid moth (*Ectropis obliqua hypulina*) was reported (*Liao et al., 2019*). In addition, we found that the levels of JA, JA-Ile, and JA-Val in the walnut husks infested by *C. pomonella* gradually increased during the infestation process. However, in tea trees, JA levels decrease with time due to damage caused by the green tea leafhopper (*Empoasca onukii*) (*Qiao et al., 2023*). We speculated that the phytohormone levels vary in different plant responses to insect pests and that in the defence of walnut plants against *C. pomonella*, SA, ABA and MeJA may participate in the initial stress response, and JA, JA-Ile, and JA-Val may play a role in long-term and sustained responses. The temporal variation in the levels of different phytohormones detected in walnut plants after *C. pomonella* infection suggested a possible synergistic or antagonistic effect on defence hormone activation.

The variation in phytohormone levels during plant defence against insects might be related to the activation of various defence genes and the production of direct defence and indirect defence compounds (*Verma, Ravindran & Kumar, 2016*; *Li et al., 2022*). The transcriptional response of plants to insect attack involves dynamic, complex reprogramming of signal transduction and biosynthesis that changes over time (*Ye et al., 2021*; *Kant et al., 2004*). SA is produced in plants through two metabolic pathways: the isochorismate (ICS) pathway and the phenylalanine ammonia-lyase (PAL) pathway (*Spoel & Dong, 2024*; *Mishra & Baek, 2021*). In this study, the transcriptomic analysis revealed that all the *PAL* genes were upregulated after *C. pomonella* damage, suggesting that the production of SA in walnuts after *C. pomonella* infestation might occur through the PAL pathway. Similarly, in a study of maize leaves after Asian corn borer infestation, all *PAL* genes were significantly upregulated, whereas the expression of *ICS* genes was downregulated (*Guo et al., 2019*). We speculated that *PAL* genes were upregulated after *C. pomonella* damage, suggesting that the production of SA in walnuts is a response to *C. pomonella* infestation and needs to be further investigated. Abscisic acid (ABA) plays an important role in drought, salt, pest, and injury responses (*Chen et al., 2020*). The populations of the aphid *Myzus persicae* were found to be smaller on the Arabidopsis ABA-deficient *aba1-1* mutant than on the Arabidopsis wild type (*Hillwig et al., 2016*). The genes involved in the ABA signalling pathway (NCED and ABR) were found to be upregulated in aphid-infested alfalfa (*Medicago truncatula*), resulting in stomatal closure and reduced leaf transpiration (*Sun et al., 2015*). However, the expression of *NCED3A*, *NCED3B*, *NCED6A*, and *NCED9A* were down-regulated in bananas treated with *Fusarium oxysporum* race 4 (*Zeng et al., 2023*). In this study, the expression of NCED in walnut husks was downregulated after different time of infestation by *C. pomonella*. Therefore, we speculated that this result may be caused by secondary infection of other microorganisms in the environment after the walnut husk was damaged by *C. pomonella*. JA also participates in regulating plant defence against herbivorous insect attacks (*Ruan et al., 2019*). Previous studies have shown that genes related to JA biosynthesis (such as *LOX*, *AOS*, *OPR*, and *JAR*) were significantly upregulated in maize leaves after infection with the Asian corn borer (*Guo et al., 2019*). In this study, the expression of key genes involved in JA biosynthesis and signal transduction (*LOX*, *AOS*, *AOC*, *OPR*, *JAZ*, and *MYC2*) in walnut plants was significantly upregulated and continued to increase until 72 h after injury, and JA accumulation increased during cooccurring moth infection, suggesting that JA plays a role in regulating walnut defence against infestation by *C. pomonella*. The transcriptional dynamics of *C. pomonella* fed field-grown walnut husks require further study. In brief, the SA, ABA, and JA genes in the synthetic pathway respond to *C. pomonella* infestation, indicating that the plant response to insect damage is a complex regulatory mechanism involving receptors that receive phagocytosis signals and internal signal transduction and response. Phytohormones, important signalling substances in plants, play diverse roles, including the regulation of insect resistance and the control of disease caused by insect damage.

Flavonoids protect plants from pathogen and insect infestation (*Ramaroson et al., 2022*; *Yang et al., 2023*). The biosynthesis and biological functions of flavonoids have been reported. *L*-Phenylalanine to 4-coumaroyl-CoA, which is catalysed by PAL, 4CL and cinnamic acid 4-hydroxylase (C4H); 4-coumaroyl-CoA to flavonone, which is catalysed by CHS and CHI; and flavonones to various types of flavonoids, which are the three stages of flavonoid biosynthesis (*Shen et al., 2022*). In the present study, a total of eight genes involved in flavonoid synthesis were significantly upregulated after *C. pomonella* infestation. These results suggest that the flavonoid pathway might be involved in the protection of walnut husks from infestation by *C. pomonella* in the natural environment.

The WGCNA results revealed a series of transcription factors related to the walnut husk defence response to *C. pomonella* infestation. Seven transcription factors (AIL, MYB, PRE, WRKY, and ERF) were identified as hub genes in the SA-related module. WRKYs have been reported to promote the accumulation of SA and to trigger the early defence response of tea plants against tea green leafhoppers (*Yang et al., 2019*). The levels of JA, JA-Ile, and JA-Val in the same module changed after *C. pomonella* infestation. *ERF27*, *bHLH35*, *bHLH18*, *TIFY5A*, *WRKY31*, and *MYB44* were identified as hub genes in the magenta module. JA biosynthesis has been reported to be regulated by a series of TFs, including MYC, MYB, ERF, WRKY, and NAC, at the transcriptional level (*Ruan et al., 2019*). Several JAZ-interacting transcription factors, including MYC, bHLH, and the WD-repeat/bHLH/MYB complex, are involved in plant-insect resistance (*Campos, Kang & Howe, 2014*; *Howe, Major & Koo, 2018*). Similarly, the levels of ABA and MeJA exhibited similar changes after *C. pomonella* infestation, and they were in the same coexpression network. Five transcription factors (bHLH, PCL, APR, bZIP, and ILR) were identified as hub genes in the green module. The bZIP transcription factor family has been reported to play a crucial role in the regulation of ABA-mediated phytohormone signalling (*Zong et al., 2016*). The genes in the coexpression module related to JA, JA-ile, JA-val, SA, ABA and MeJA identified in this study may play an important roles in regulating the response of walnut husk to damage caused by the moth. However, the molecular functions of these genes in walnut plants require further investigation.

In summary, in this study, the resistance of walnut husks to *C. pomonella* larvae were investigated, the transcriptional dynamics underlying the defence response of walnut husks against *C. pomonella* larvae were analysed at different time points, and the roles of the JA, SA, and ABA signalling pathways as well as flavonoid synthesis pathways in the defence response were highlighted. The hub genes involved in the SA-, ABA-, JA-, JA-Ile-, JA-Val-, and MEJA-mediated defence responses of walnut husks to *C. pomonella* were screened *via* the construction of a coexpression network. This study laid the foundation for the selection of resistant walnut varieties and the ecological control of *C. pomonella*. In follow-up studies, the role of these hub genes in the walnut husk defence mechanism against *C. pomonella* will be further identified.

## CONCLUSIONS

By combining transcriptome and hormone analyses, this study revealed the defensive effects of walnut husks on combating moth damage. Our results suggest the following conclusions: (i) The infestation of *C. pomonella* at different time points induced a large number of DEGs in walnut husks, with the expression levels of these DEGs showing different trends within 0–72 h of *C. pomonella* damage. (ii) After different durations of *C. pomonella* infestation, DEGs involved in biosynthesis of secondary metabolites, metabolic pathways, alpha-linolenic acid metabolism, and phenylpropanoid biosynthesis were produced, indicating that the resistance of walnut husks activated by *C. pomonella* infestation was controlled by multiple plant stress resistance pathways. (iii) Genes closely related to phytohormone synthesis (SA, ABA, and JA) and flavonoid biosynthesis, which were crucial for insect resistance in walnut husks, were strongly expressed during *C. pomonella* infestation, suggesting that the SA, ABA, and JA signalling pathways, along with the flavonoid biosynthesis pathway, were involved in the resistance response of walnut husks induced by *C. pomonella* infestation. Taken together, our study reveals the defence response of walnut husks to *C. pomonella* infestation, providing a basis for the breeding of insect-resistant walnut varieties.

### Funding

This study was funded by the Key Research and Development Projects in the Xinjiang Uygur Autonomous Region (2021B02004). The funders had no role in study design, data collection and analysis, decision to publish, or preparation of the manuscript.

### Grant Disclosures

The following grant information was disclosed by the authors:
Key Research and Development Projects in the Xinjiang Uygur Autonomous Region: 2021B02004.

### Competing Interests

The authors declare that they have no competing interests.

### Author Contributions

- Xiaoyan Cao conceived and designed the experiments, performed the experiments, analyzed the data, prepared figures and/or tables, authored or reviewed drafts of the article, and approved the final draft.
- Xiaoqin Ye performed the experiments, prepared figures and/or tables, and approved the final draft.
- Adil Sattar conceived and designed the experiments, authored or reviewed drafts of the article, and approved the final draft.

## DNA Deposition

The following information was supplied regarding the deposition of DNA sequences:

The data are available at GenBank: PRJNA1140835.

## Data Availability

The data are available in the Supplemental File.

## Supplemental Information

Supplemental information for this article can be found online at http://dx.doi.org/10.7717/peerj.18130#supplemental-information.

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
