# Peer review of "Transcriptomic and coexpression network analyses revealed the regulatory mechanism of Cydia pomonella infestation on the synthesis of phytohormones in walnut husks"

_PeerJ, doi:10.7717/peerj.18130_

## Round 0.1 · original submission · Major Revisions

· Academic Editor

Major Revisions

Authors should respond to the criticism of the reviewers, in particular reviewer 1.

Reviewer 1 ·

Basic reporting

I read this article with interest. Authors investigated and analyzed walnut husk to feed on codling moth and sampled walnut exocarp after codling moth feeding, analyzed the hormonal changes and sequenced the transcriptome, and mined key genes in several modules by combining the hormonal and transcriptome data through the WGCNA.

Experimental design

1. In terms of the logic of the article, result 1 does not seem to be related to the following. This part proves that feeding walnut husk can reduce C. pomonella damage to the walnuts by inhibiting the growth of C. pomonella, and according to the normal logic, the cause of slow growth of codling moth should be further analyzed.
2. Previous studies have shown hormone and flavonoid content change when plants are infested by pests. This process is also emphasized in this paper, but the changes in flavonoids after infestation were not measured and analyzed, and it is recommended that this section be added.
3. In this manuscript, uninfected walnut husks were collected as control, why not use artificially damaged green peels as the control.
4.Phytohormones, especially jasmonate (Abraham J.K. Koo, 2009) and the phenylpropane pathway (Mayu Yoshikawa, 2018) are altered when plants are injured (injury by herbivores, pathogens, mechanical stress, and other environmental insults). Although this MS confirms that the metabolism of jasmonate and flavonoids is altered when the husk are infected by codling moths, and that walnut husks reduced C. pomonella damage to the walnuts by inhibiting the growth of C. pomonella., it does not seem to have further resolved the intrinsic causes.

Validity of the findings

The manuscript used weighted gene co-expression network analysis to obtain many key transcription factors, which seems to have achieved certain results, and seems to have no substantial progress, unable to identify the key genes, and lack of follow-up in-depth research.

Additional comments

Description of applied methods and used materials is in part superficial. Pictures of the sampling method should be added, such as the sample site and size.

Reviewer 2 ·

Basic reporting

The language expression of this article is clear, the background description is clear, the structure design is reasonable. But no appropriate raw data been made available.

Experimental design

This approach is highly intriguing, sound, and the figures are clearly presented.

Validity of the findings

The subject matter at hand is truly innovative, especially when considering the limited existing literature in this particular field.

Additional comments

1. Line 118: The first group was given walnut husks, while the second and third groups were provided with fed kernels. Was there a specific reason for not administering equal amounts?

2. Line 127: Please provide us with a summary of 4th instar C. pomonella larvae.

3. Line 218: The content is quantified, while the figure indicates the concentration.

4. Line 219: “There were no changes in the levels of H2JA, OPC-4, OPDA, or
SAG at any time during C. pomonella infestation.”However, this method was not observed for the determination of several substances.

5. Line 262: Could you provide an explanation for the down-regulation of NCED genes?

6. Line 292: The author examines all modules with positive correlation, but why not also consider the modules with negative correlation?

7. Line 323: In the turquoise module, the range of upregulation levels was greatest at 24 h after infestation; in the green module, the expression level was significantly up regulated at 48 h after C. pomonella infestation; in the magenta module, the expression level was significantly up regulated at 72 h. Should there be a module of genes that exhibit significant up regulation at 12 hours?

8. Line 395: “We speculated hysteresis may occurduring the transduction of ABA signals after C. pomonella infestation, but ABA synthesis is not affected.”What does that mean? What evidence exists to substantiate the author's hypothesis?

9. Line 462: The information and format of some of the cited references are problematic. Such as Line 536, it should be “The plant journal”, not “The plant journal for cell and molecular biology”; “The walnut (Juglans regia)” should be “The walnut (Juglans regia)”. Please meticulously verify all references for accurate information and proper formatting.

10. The information presented in Figure 7 is overly simplistic. It is requested to enhance the level of detail for each diagram.
11. The discussion section lacks sufficient references to recent relevant literature, which should be included.

Reviewer 3 ·

Basic reporting

According to the above review comments, the author is suggested to make corresponding amendments and improvements. After the author fully responded to the review comments and made necessary amendments to the paper, the reviewer recommended the paper for publication.

Experimental design

The Materials and Methods section requires a description of the specific location where the test material was grown, and an overview of the test site.
The methodology section is insufficiently detailed, making it difficult for readers to evaluate the research process and replicate the study. Please provide more information on the experimental design, data collection procedures, and statistical analysis.

Validity of the findings

no comment

Additional comments

This paper is to study the key genes of disease resistance induced by endogenous hormones in walnut shell, but the content of this paper focuses on the analysis of expression pathways of genes related to ABA, SA and JA biosynthesis and signal transduction, and the conclusion is not clear about what the key genes are. Therefore, the title of this article is inappropriate and does not match the research content, and it is suggested to modify it.

Annotated reviews are not available for download in order to protect the identity of reviewers who chose to remain anonymous.

·

Basic reporting

Very well written. My only comment for improvement is that figure 3 is very busy. Suggestions: Show one pathway per graph with different time points so that any changes can be observed. Only need the key in one place rather than on each graph, as it detracts from viewing the data.

Experimental design

No comment.

Validity of the findings

No comment.

Additional comments

This is a solid paper with very clear data, well described context, and a path forward for future investigations.

---

## Round 0.2 · Minor Revisions

· Academic Editor

Minor Revisions

I agree with the reviewer, the conclusions should be more focused

Reviewer 1 ·

Basic reporting

The conclusion section needs to be rewritten to give clear conclusions, such as those with metabolic pathways involved in defense processes, rather than just those with plant hormones.

Experimental design

-

Validity of the findings

-

Additional comments

-

---

## Round 0.3 · accepted · Accept

· Academic Editor

Accept

Now MS can be accepted, authors replied satisfactorily to reviewer 1